# Effect of Fish Stock Density on Hormone Genes Expression from Brain and Gastrointestinal Tract of *Salmo salar*

**DOI:** 10.3390/ani12091174

**Published:** 2022-05-04

**Authors:** Claudio A. Álvarez, Paula A. Santana, Claudia B. Cárcamo, Constanza Cárdenas, Byron Morales-Lange, Felipe Ramírez, Cristian Valenzuela, Sebastián Boltaña, Javier Alcaíno, Fanny Guzmán, Luis Mercado

**Affiliations:** 1Laboratorio de Fisiología y Genética Marina, Centro de Estudios Avanzados en Zonas Áridas, Coquimbo 1781421, Chile; claudia.carcamo@ucn.cl; 2Facultad de Ciencias del Mar, Universidad Católica del Norte, Coquimbo 1781421, Chile; 3Instituto de Ciencias Químicas Aplicadas, Facultad de Ingeniería, Universidad Autónoma de Chile, San Miguel, Santiago 8910060, Chile; paula.santana@uautonoma.cl; 4Centro de Innovación Acuícola, Aquapacifico, Coquimbo 1781421, Chile; 5Núcleo Biotecnología Curauma (NBC), Pontificia Universidad Católica de Valparaíso, Valparaíso 2374631, Chile; constanza.cardenas@pucv.cl (C.C.); fanny.guzman@pucv.cl (F.G.); 6Grupo de Marcadores Inmunológicos, Laboratorio de Genética e Inmunología Molecular, Instituto de Biología, Pontificia Universidad Católica de Valparaíso, Valparaíso 2374631, Chile; byron.morales@pucv.cl (B.M.-L.); f.c.ramirez.cepeda@gmail.com (F.R.); cristian.valenzuela@pucv.cl (C.V.); 7Department of Oceanography, University of Concepción, Concepción 4070386, Chile; sboltana@udec.cl; 8AquaAdvise-Fundación Chile, Puerto Montt 5480000, Chile; javier.alcaino@fch.cl

**Keywords:** stress, peptide hormone, stock density, *Salmo salar*

## Abstract

**Simple Summary:**

Various long-term stress conditions may exist in fish cultivation, damaging the physiological responses that regulate the fish growth and feed. Different signalers connect the brain with the gastrointestinal tract, including the perception of stress factors for the regulation of physiological responses. Here, we evaluated the effect of varying culture densities of *Salmo salar* post-smolt on the gene expression of some brain and gastrointestinal hormone signalers. We found that high stock densities could promote the levels of molecules associated with feed inhibition, which could be related to the stress pathway regulated by corticoids. Thus, the expression of these peptide hormones could be used as biological markers to improve production practices in fish aquaculture.

**Abstract:**

A variety of long-term stress conditions may exist in fish cultivation, some of which are so severe that fish can no longer reestablish homeostasis. In teleost fish, the brain and gastrointestinal tract integrate signals that include the perception of stress factors regulating physiological responses, such as social stress by fish population density, where peripheral and central signals, such as peptide hormones, are the main regulators. Therefore, we proposed in this study to analyze the effect of different stock densities (SD) in the gene expression of brain neuropeptide Y (NPY) and calcitonin gene-related peptide (CGRP), together with the gastrointestinal peptide hormones leptin (Lep), vasointestinal peptide (VIP), and protachykinin-1 (Prk-1) in *Salmo salar* post-smolt. The coding sequence of *S. salar* VIP and Prk-1 precursors were firstly cloned and characterized. Then, the mRNA expression of these genes, together with the NPY, Lep, and CGRP genes, were evaluated in post-smolts kept at 11 Kg/m^3^, 20 Kg/m^3^, and 40 Kg/m^3^. At 14 days of culture, the brain CGRP and liver leptin mRNA levels increased three and tenfold in the post-smolt salmons kept at the highest SD, respectively. The high levels of leptin were kept during all the fish culture experiments. In addition, the highest expression of intestine VIP mRNA was obtained on Day 21 in the group of 40 Kg/m^3^ returning to baseline on Day 40. In terms of stress biochemical parameters, cortisol levels were increased in the 20 Kg/m^3^ and 40 Kg/m^3^ groups on Day 40 and were the highest in the 20 Kg/m^3^ group on Day 14. This study provides new insight into the gastrointestinal signals that could be affected by chronic stress induced by high stock density in fish farming. Thus, the expression of these peptide hormones could be used as molecular markers to improve production practices in fish aquaculture.

## 1. Introduction

The welfare of farmed fish has become a vital issue for their successful intensive production. In this way, the adequate evaluation of effective procedures and management of the physiological processes of these animals is essential to give recommendations that improve future practices and legislation in this area [1]. Therefore, it is necessary to understand how different neuroendocrine and metabolic mechanisms in aquaculture species can be affected by production practices.

Salmon aquaculture manages large populations of organisms in limited spaces. Thus, it is necessary to study the effect these stock densities have on different aspects of productive importance, such as growth rates or feed conversion ratios. The alterations of these performance indicators in farmed fish have been attributed to a stress condition [2,3,4], which occurs when an animal cannot maintain a normal physiological state due to various stressors that negatively affect its well-being. These sets of physiological responses are jointly regulated by stress hormones in teleost fish, namely catecholamines and cortisol [5,6]. The changes promoted by these hormones can be summarized in two fundamental aspects: (1) the mobilization of energy substrates and (2) the modifications of the circulatory, respiratory, and osmoregulation systems. The fast alteration of these responses is required to overcome the stressor; however, different physiological processes such as immune response and somatic growth are interrupted if the stressor is maintained over time. Consequently, both animal growth and welfare are affected [7].

A generic response of animal behavior to stressful situations is the reduction of food intake; as a result, both muscle mass and fish size are reduced. It has been shown that various types of stressors, including physicochemical changes in water, the presence of pathogenic microorganisms, and non-standardized stock densities, inhibit intake and compromise the growth in various species of fish, such as Atlantic salmon, Senegalese sole, and goldfish [8,9,10,11,12]. Furthermore, various studies describe that chronic stress situations cause reduced or even negative growth rates or decreased feed conversion efficiency [7,12,13,14]. Thus, the effect of stress on food intake results in growth limitations, which is probably one of the main factors affecting fish farming productivity.

The molecular mechanism involved in regulating food intake in non-mammals is still weak. However, different studies in teleost fish suggest that it works similarly in fish and mammals [15]. Food intake is a complex process coordinated by specific areas of the hypothalamus. However, it requires peripheral signals from various organs, including the gastrointestinal tract and the liver [16]. The signals can be peptides or low-molecular-weight proteins that can increase or decrease food intake, described as orexigenic or anorexigenic factors, respectively [17]. In fish, some of these molecules have been identified as homologous to those already described in mammals. The effects of these peptides in both feeding and energy metabolism have been proved by intraperitoneal or intracerebral injection [15]. Furthermore, the mRNA expression of orexigenic or anorexigenic genes has been used to determine the effects of different culture conditions, such as photoperiod, physicochemical parameters of water condition, and different feeding conditions [18,19,20].

Different studies conducted on Teleostei demonstrate that peptide hormones such as leptin (Lep) can be critical in the anorexigenic effects produced under stress. In teleost fish, the leptin mRNA is synthesized mainly in the liver, and direct association with cortisol pathways activation has been described in different species, such as rainbow trout and goldfish [21,22]. Furthermore, the exogenous application of Lep can suppress food intake in Atlantic salmon and rainbow trout [23,24]. These observations were accompanied by results at a transcriptional level that display a marked reduction in the expression of orexigenic peptides such as neuropeptide Y (NPY) [23,24,25,26].

The presence of food in the gastrointestinal tract plays a critical role in feeding regulation, where orexigenic or anorexigenic peptides are involved in the control of both intestinal motility and secretion [27]. In addition, some of these peptides are part of the peripheral satiety system transmitting these signals to the hypothalamus to stop the food intake. Homologous brain-gut peptides of mammals have been described in teleost fish, mainly as anorexigenic peptides such as vasoactive intestinal peptide (VIP), calcitonin gene-related peptide (CGRP), and tachykinins [28,29,30]. However, few studies have evaluated how different stressors affect the gut–brain neuroendocrine signaling in fish. In fact, the identification or characterization of some of these genes in Atlantic salmon, such as VIP and tachykinins, has not been described yet.

Overcrowding in fish farming could activate chronic stress conditions affecting physiological responses associated with brain–gastrointestinal communication. For that reason, in this work, we aimed to evaluate the expression of the peptide hormone genes of these physiological pathways in Atlantic salmon post-smolts that maintained 11 Kg/m^3^, medium 20 Kg/m^3^, and 40 Kg/m^3^ under recirculating aquaculture system. First, the genes of gastrointestinal peptide hormones not described yet for Atlantic salmon, VIP precursor, and protachykinin-1 (TAC-1), were cloned and characterized. Then, the mRNA expression of these genes, together with NPY, Lep, and CGRP, were evaluated in post-smolts kept at different densities. For the evaluation of possible chronic stress conditions, the relative expression of these genes was determined by RT-qPCR after 14, 21, and 40 days of culture. Moreover, plasma cortisol levels were evaluated at each sampling, and putative transcription regulatory elements associated with glucocorticoid were searched in the peptide hormone genes. This study provides new shreds of evidence in favor of sustaining that gastrointestinal signals are affected by chronic stress induced by high stock density in fish farming.

## 2. Materials and Methods

### 2.1. Fish Maintenance and Sampling

Post-smolts of *Salmo salar* (n = 930; 220 g) of one batch of fish from the same production cycle were maintained in Fundación Chile facilities in Puerto Montt, Chile. Fish were held at 15 Kg/m^3^ in circular fiberglass tanks of 350 L equipped with oxygen pumps, biological filters, ultraviolet sterilizing units, and flow-through seawater (salinity at 32%). After the adaptation period, salmon post-smolts were randomly distributed into 12 circular fiberglass tanks of 350 L (four at each stocking density level). Different numbers of fish were added to each tank to obtain the three stocking densities of 11 Kg/m^3^ (20 fish per tank), 20 Kg/m^3^ (35 fish per tank), and 40 Kg/m^3^ (70 fish per tank). To facilitate the description throughout this work, they will be referred to as low stocking density (LD), medium stocking density (MD), and high stocking density (HD), respectively. Fish were fed to satiety with commercial dry feed pellets daily (Skretting). Oxygen saturation (102.9 ± 3.9%) and temperature (9.7 ± 1.3 °C) were tested daily during the experimental period. Fish were held under artificial photoperiod of 24L:0D.

Fish were maintained at different stock densities for 14 days as an acclimation time. Therefore, the first sampling day was performed on Day 14 of culture. Then, another two sampling points were carried out to analyze the chronic effect of stock density on Days 21 and 40. Fish were fasting one day before the sampling day. Three fishes from each tank (12 from each experimental group) were randomly collected to evaluate biochemical blood parameters and gene expression analysis in different tissues. Sampled fish were captured and quickly anesthetized with AQUI-S^®^ (13.3 mg/L, isoeugenol 50%, Bayer-Germany). Fish were maintained and handled following the guidelines of experimental procedures in animal research from the Ethics and Animal Welfare of the Chilean National Agency for Research and Development (ANID-Chile).

Total length (TL) was measured to the nearest millimeter from the tip of the snout to the end of the middle caudal-fin rays. The total weight (TW) was determined as the total body weight with a digital balance to an accuracy of 0.01 g.

Blood samples were taken from the caudal vein using heparinized syringes. Blood samples were centrifuged immediately after collection at 5000× *g* for 5 min at 4 °C, and the plasma fraction was stored at −20 °C. After blood collection, fish were decapitated.

The tissue samples (liver, brain, and gut) were placed in sterile tubes and immediately stored in 400 µL of RNAlater (RNA Stabilization Reagent, Thermo Fisher).

### 2.2. Biochemical Assays

Glucose concentration from plasma was measured using commercial kits from Wiener Lab (Brazil) following the manufacturer’s protocols. Cortisol concentration from plasma was measured using a colorimetric competitive enzyme immunoassay (Cortisol EIA kit) following the manufacturer’s protocols.

### 2.3. Total RNA Extraction and cDNA Synthesis

Total RNA of *S. salar* tissue was extracted from tissues using E.Z.N.A.^®^ Total RNA Kit II (Omega Bio-tek, Norcross, GA, USA) according to the manufacturer’s protocols. RNA was then treated with DNAse I (Promega, Madison, WI, USA) for 15 min at room temperature and inactivated for 10 min at 65 °C. Next, total RNA was quantified with an Epoch spectrophotometer (BioTek, Winooski, VT, USA). The quality was determined by agarose gel electrophoresis. Finally, cDNA synthesis was carried out using PrimeScript™ RT Reagent Kit with gDNA Eraser (TaKaRa Bio-INC, Shiga, Japan) and oligo-p(dT)_15_ primer following the manufacturer’s protocols.

### 2.4. Cloning and Bioinformatics Analysis of Nucleotide Sequences

The complete coding sequences (CDS) of ssVIP and SsTAC-1 were obtained by an EST contigs sequences search from *S. salar* in the GenBank database. Primers designed for the amplification of these genes are listed in Appendix A. Liver and intestine cDNA templates were used to amplify SsTAC-1 and SsVIP, respectively. The PCR product was then purified using an E.Z.N.A.^®^ Gel Extraction Kit (Omega Bio-tek, Norcross, GA, USA). The amplicon sequence was verified by sequencing (Macrogen Inc., Seoul, Korea).

The putative protein sequence was obtained from the Expasy portal (http://www.expasy.org/, accessed on 17 November 2021). Alignment of multiple sequences was performed using the ClustalW tools in MEGA 6.0 software. The sequences and percent identity matrix for TAC1 and VIP protein precursors alignment are listed in Appendix A, respectively.

Tertiary structure models of putative mature peptides from Atlantic salmon TAC1 and VIP protein precursor were analyzed in the PEP-FOLD3 web server (http://bioserv.rpbs.univ-paris-diderot.fr/services/PEP-FOLD3, accessed on 22 November 2021) [31]. After choosing the best model, the 3D structures of peptides were constructed using PyMOL.

Genomic sequences for the leptin and VIP mRNA of Atlantic salmon were retrieved from NCBI databases, including 5 and 3’ UTRs and feature annotations. First, the sequences were aligned in CLC Main Workbench 6.9.2 (https://digitalinsights.qiagen.com, accessed on 15 December 2021). Then, the 5′UTR region was analyzed through motif-based sequence analysis tools. First, MEME suite [32] was used to search for motifs, and then STAMP [33] was used to classify these motifs according to the Jaspar database [34]. In the same way, the motifs related to steroid hormone receptor were searched in the Jaspar database, and then using Find Individual Motif Occurrences (FIMO) [35] from the MEME suite, these motifs were searched in the 5′ UTR sequences. Finally, all the discovered motifs were annotated in the genomic sequences, and the weblogos were generated at https://weblogo.berkeley.edu (accessed on 16 November 2021) [36].

### 2.5. Gene Expression Analysis by RT-qPCR

Specific primers were designed to amplify appetite-regulating genes for *S. salar* (Appendix A). In order to validate the elongation factor 1 alpha (EF-1α) as a housekeeping gene for the samples, statistical tests on EF-1α expression values among different tissues or stress conditions were performed. Non-significant differences were found among them (*p* > 0.05). RT-qPCR was performed using 20 µL reaction mixtures containing Maxima^®^ SYBR Green/ROX qPCR Master Mix (Thermo Scientific, Rockford, IL, USA), 0.3 μM (final concentration) of each primer, and 2 µL of cDNA. Primer pair efficiencies (E) were calculated from the given slopes according to the equation: E = 10[−1/slope] (Appendix A). Assays were carried out in an Mx3000P qPCR System (Agilent Technologies, Santa Clara, CA, USA) with an initial denaturation step of 10 min at 95 °C, followed by 40 PCR cycles of denaturation step (95 °C, 15 s) and annealing–extension step (60 °C, 1 min). Finally, the melting curve was obtained at 75–95 °C with a heating rate of 0.1 °C per second and continuous fluorescence measurement. Relative expression was calculated using the −2ΔΔCq method [37] using the measured quantification cycle (Cq) values of the EF-1α housekeeping gene to normalize the measured Cq values of target genes.

### 2.6. Statistical Analysis

Acclimation tanks were analyzed for a tank effect using a one-way analysis of variance (ANOVA). No differences were found between different experimental tanks, so individual fish were considered as replicates in the analyses. In order to determine statistical differences among treatments, the normality and homogeneity of variance of the data were analyzed by the Shapiro–Wilk test and Levene’s test. Then, one-way ANOVAs were used to compare gene expression and stock density. Finally, two-way ANOVAs were used to compare gene expression concerning different times and densities. Data were analyzed using R version 3.5.2 software. Differences were considered significant when *p* < 0.01 (**) or *p* < 0.05 (*). Results are represented graphically as the mean values ± standard deviation (SD) with GraphPad Prism 6.0 software.

## 3. Results

### 3.1. Characterization of the Coding Sequence of TAC1 and VIP of S. salar

The salmon TAC1 and VIP coding sequences (CDS) were completely sequenced, obtaining the putative amino acid composition of the protein precursors (GenBank Z019485). The prepro-tachykinin of *S. salar* (ssPrk1) from TAC1 CDS was 113-aa in length, including an 18-aa signal peptide (Figure 1A). The amino acid alignment among ssPrk1 protein sequences with homologous teleost fish revealed high sequence similarity in different regions, mainly localized mature peptides (Figure 1, Appendix A). Two peptides named Substance P (SP) and Neurokinin A (NKA) were delimited by cleavage sequences at the N-terminal end of ssPrk1 (^55^RK^56^ and ^79^KR^80^, respectively). In addition, the C-terminal end of both ssSP and ssNKA peptide sequences shows a glycine after the cleavage site, which indicates an amidation site (Figure 1A). The high sequence similarity with human peptides allowed the prediction of high-quality structures of ssSP and ssNKA by using crystal structures previously reported for both peptides of *Homo sapiens* (PDB ID 2KS9 and 1P9F, respectively). These peptides with only 10–11 residues in length can form putative short α-helices as secondary structures (Figure 1B,C).

Conversely, the coding sequence of *S. salar* VIP gave a predicted prepro-VIP with 151 amino acid residues in length (GenBank MZ019486). The cleavage sites display a remarkable homology with the VIP precursor of teleost fish (Figure 2A, Appendix A), resulting in two mature peptides excised from the ssVIP precursor. In teleost fish, these peptides are named peptide-histidine-isoleucine (PHI) and vasoactive intestinal peptide (VIP). Similar to mature peptides cleaved from TAC1, the presence of a glycine residue after the cleavage site in the C-terminal end of both VIP and PHI peptides indicates a putative amidation site (Figure 2A). Moreover, the predicted peptides yielded a high-resolution model using as a template the crystal structure obtained for VIP of *H. sapiens* (PDB ID 2RRH), observing an α-helix with very defined features, including the thickness of the coil and the length of each complete turn along the helix axis (Figure 2B,C).

### 3.2. Growth Performance and Biochemical Parameters of Post-Smolt S. salar Kept at Different Stock Densities

Post-smolts of *S. salar* were kept at three different stock densities: 11, 20, and 40 Kg/m^3^. Three samplings were carried out throughout the bioassay at 14, 21, and 40 days of culture.

At the end of the experiment, a significant increase in both total weight and total length was observed in all-stock densities tested (*p* < 0.05) (Figure 3). There were no significant differences in final weights and final lengths (*p* < 0.05) among the stocked fish across the three treatments regarding sampling day.

Plasma cortisol and glucose were evaluated as chronic stress indicators. The fish kept at 20 Kg/m^3^ showed the highest plasma cortisol values in the three days of sampling (*p* < 0.05), (Figure 4A). It is important to note that the groups of fish kept at 20 Kg/m^3^ exhibited a wide range in magnitude of cortisol levels on Days 14 and 21, similarly to fish kept at 40 Kg/m^3^ on Day 40.

The concentration of plasma glucose was significantly lower on Day 21 in both fish kept at 20 Kg/m^3^ and 40 Kg/m^3^ (*p* < 0.05) (Figure 4B). Moreover, a significant decrease in plasmatic glucose in the group of fish kept at the lowest density was observed on Day 40 of culture (*p* < 0.05).

### 3.3. Effects of the Stock Density on Gene Expression of Peptide Hormones of Brain and Gastrointestinal Tract

The transcriptional expressions of genes related to central and peripheral signals of food intake were evaluated in smolt salmons kept at different stock densities and expressed as fold change with respect to fish kept at 11 Kg/m^3^. At 14 days of culture, the liver leptin mRNA levels increased 10-fold in the post-smolt salmon kept at the highest stock density. These levels were kept during the next sampling days (Figure 5A). Nevertheless, the groups of fish kept at 20 and 40 Kg/m^3^ exhibit a wide range in the magnitude of gene expression of leptin on Day 40. The latter was also observed in the group of fish kept at 20 Kg/m^3^ after 40 days of culture.

In the case of intestine VIP mRNA levels, the highest expression on Day 21 was obtained in the group of 40 Kg/m^3^, returning to baseline on Day 40 (Figure 5C). On the other hand, the mRNA expression of liver TAC-1 and brain NPY did not show significant differences among the different stocking densities during the bioassay days (Figure 5A,B). Again, it is important to note that groups of fish kept at 40 Kg/m^3^ exhibit a wide range in the magnitude of gene expression of CGRP after 14 and 21 days of culture (Figure 5B).

### 3.4. Identification of DNA Sequence Recognition by Steroid Hormones Receptors in Leptin and VIP Genes

The exon–intron structure of the Lep and VIP genes is shown in Figure 6. Both leptin and VIP genomic sequences exhibit sequences of motifs from the steroid hormone receptor family. In the case of leptin, there are two superpose motifs NR3C1 (MA0113.2) and ESR2 (MA0258.1) (Figure 6A); and in the VIP sequence, five putative motifs were found: NR3C1, NR3C2 (MA0727.1), SRF (MA0083.1), ESR1 (MA0112.3), and Esrrg (MA0643.1) (Figure 6B).

## 4. Discussion

The sustainability of the aquaculture industry relies on fish health, welfare, and productivity. Therefore, optimizing the culture parameters, one of which is the stocking density, can prevent damage to several physiological processes in the farm fish and ensure optimal growth performance. Here, we describe the physiological effects of different initial stock densities of *S. salar* post-smolts cultured under a recirculating aquaculture system (RAS). At the end of the experiment, which lasted 40 days after transference of *S. salar* post-smolts to the three different stocking densities (11, 20, and 40 Kg/m^3^), no significant differences were observed both in body weight and total length. However, there was a trend toward higher body weight in the lowest and medium stocking density groups. These results are in concordance with previous studies [38,39,40,41], where the authors described a significantly higher specific growth rate and body weight gain in post-smolts *S. salar* salmon kept at a stock density lower than 30 Kg/m^3^. Therefore, stocking density had directly adverse effects on fish growth.

The effect of stocking density on growth performance in teleost fishes depends on the physiological conditions of the fish and on the conditions of the environment where they develop (Álvarez et al., 2020; Refaey et al., 2018). Thus, the physiological effects of this parameter are species-specific. A non-standardized stocking density is associated with increasing size heterogeneity related to modification of social interactions, involving activation of stress routes in the farm fish species (Álvarez et al., 2020; Manley et al., 2014). Interestingly, in the present study, the highest cortisol level was observed in the group kept at 20 Kg/m^3^ and remained high throughout the bioassay, while fish kept at the highest density tested began to increase their cortisol levels on Day 40. According to the study by Wang et al. [41], the cortisol levels of *S. salar* were increased after 40 days of culture in the group of fish kept at densities above 15 Kg/m^3^. In addition, similar to our study, those levels remained high until the end of the experiment. So, it can be assumed that the group of fish kept at 40 Kg/m^3^ probably continued to raise plasma cortisol levels after 40 days. Future studies should continue analyzing whether this activation of stress routes is maintained for more extended periods because adapting physiological mechanisms have been described in farm fish, such as secreting less cortisol as stocking density increases (Long et al., 2019; Paredes-López et al., 2021; Vijayan and Leatherland, 1990).

Although the increase in plasma cortisol concentration is a stress indicator most commonly used among Teleostei, the fish may still respond to a stressor despite a return of plasma cortisol to basal levels (Aerts et al., 2015; Vijayan and Leatherland, 1990). There are also reported cases where chronic cortisol elevation was not significantly produced compared to the basal level, despite subjecting fish to prolonged situations of severe stress (Barton, 2002). In such cases, a basal cortisol concentration can be misleading. Therefore, the genetic expression of biomarkers involved in the altered physiological responses under these conditions may allow determining chronic stress more accurately. Here, the focus was on the probable stress route activation by stocking density in post-smolt *S. salar*, analyzing the gene expression of peptide hormones involved in the communication of the brain with the gastrointestinal tract of Teleost fishes. In this way, we observed that gastrointestinal peptide hormones, such as vasoactive intestinal polypeptide and leptin, increased when the *S. salar* post-smolts were maintained at the highest stock density tested. Interestingly, on Day 40, the levels of ssVIP mRNA returned to basal levels, while leptin gene expression remained elevated during the experimental period. Previous studies showed that leptin is involved in mobilizing fatty acids from the gastrointestinal tract as compensation for energetically demanding stress (Copeland et al., 2011). So, probably this peripheral signal is maintained if the stressor is kept for a long time. In this way, if leptin and VIP expression remain high for a long time, it could induce unfavorable effects on growth since the fish would use their energy reserves to keep the stress response active. Thus, a possible explanation for the trend observed for lower growth at higher crop density can also be associated with the mobilization of energy sources mediated by these hormones.

In fish, as in mammals, the food intake regulation is an interconnected network from the appetite brain signals in the hypothalamus to the peripheral signals from gastrointestinal tissues, incorporating the energy status and hunger/satiety signals (Rønnestad et al., 2017). Moreover, most of those signals are evolutionarily conserved from non-mammalian vertebrates to mammals, as seen for the mature peptides of ssPrk and ssVIP precursors. The putative amino acid sequence of prepro-tachykinin and prepro-VIP of *S. salar* shows high conservation of the cleavage protease site that gives rise to the C-terminus amidated peptides from their protein precursors. Therefore, well-conserved functions would be expected; however, future studies should confirm this.

The vasoactive intestinal polypeptide (VIP) is involved mainly in smooth muscle relaxation and vasodilation of gastrointestinal tissues in high vertebrates (Bohler et al., 2021). Moreover, anorexigenic effects in mammals, birds, and teleost fish have been described (Bohler et al., 2021). In the goldfish, *Carassius auratus*, the intracerebroventricular or intraperitoneal application of VIP induced a significant decrease in food intake (Matsuda et al., 2005). However, the mechanism used by VIP to modulate this response is not well understood yet. Overall, the evidence in other vertebrates suggests that the anorexigenic effects of VIP could be mediated by the activation of corticotropin-releasing factor (CRF) neurons (Tachibana et al., 2004). CRF is one of the key factors of the hypothalamic-pituitary–interrenal (HPI) axis that ends with cortisol secretion (Lai et al., 2021). Thus, if the expression of VIP depends on HPI axis activation, the transcription of its mRNA could be regulated by the cortisol receptor. Here, the bioinformatics analysis of the *S. salar* genome allowed us to localize the partial exon–intron structure of the VIP gen, finding different putative binding sites of steroid receptors in the promotor region. Interestingly, both VIP and leptin genes contain the glucocorticoid response elements for glucocorticoid receptor NR3C1 (nuclear receptor subfamily 3 group C member 1). Therefore, the presence of these regulatory zones suggests that cortisol regulates both genes. However, future research is needed to establish if a direct relationship exists between the expression of this peptide hormone and the glucocorticoid receptor-mediated cortisol effects.

In our study, the most significant effect of the gene expression modulation by stock density was obtained in peptide hormones acting in the gastrointestinal tract of *S. salar*. However, it is also possible to observe that brain anorexigenic signals, such as CGRP in teleost (Martínez-Álvarez et al., 2009), show a tendency to increase at the highest density. Nevertheless, the CGRP gene expression in the highest stock density showed two fish groups, one with overexpression of CGRP and the other with downregulation of CGRP mRNA. This result suggests that social relationships, such as feeding hierarchies in larger groups of fish, could be further promoted at high population density, which has been previously described in salmonids (Cubitt et al., 2008; Mccarthy et al., 1992). This crowding stress leads to fish aggression and competition for food and space. Therefore, one group of fish has more difficulty in accessing feed and/or decreased appetite. In the latter case, cortisol-modulated stress could affect hormonal signals, such as leptin and VIP, by regulating their expression by means of the regulatory elements present in their mRNA sequences.

## 5. Conclusions

Social stress is important for animal welfare and especially important for animals reared in captivity, such as in salmon farming practices. Here, the data support that high stock density in Atlantic salmon post-smolt induces the overexpression of peptide hormone mRNA associated with anorexigenic signalizing. In addition, these molecules could be regulated by cortisol through glucocorticoid-responsive elements. Therefore, we propose that peptide hormones of brain–gastrointestinal tract communication could be used as a biomarker to analyze different production parameters in fish aquaculture. Future research will be performed to analyze if food intake and growth rate are associated with appetite peptide hormone functionality in *S. salar* post-smolts.

## Figures and Tables

**Figure 1 animals-12-01174-f001:**
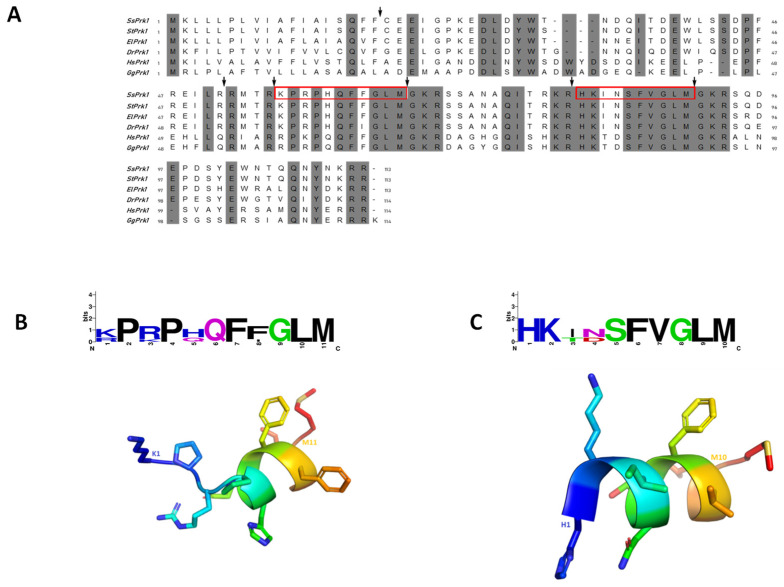
Characterization of prepro-tachykinin of *S. salar* and secondary structure modeling of mature peptides. (**A**) Amino acid sequence alignment of Prk1 of teleost species and vertebrates. The arrows indicate the putative protease cleavage, and red boxes indicate the Substance P (SP) and Neurokinin A (NKA) mature peptides. (**B**) PyMOL generated the overlapping model based on SP mature sequence. (**C**) PyMOL generated the overlapping model based on NKA mature sequence.

**Figure 2 animals-12-01174-f002:**
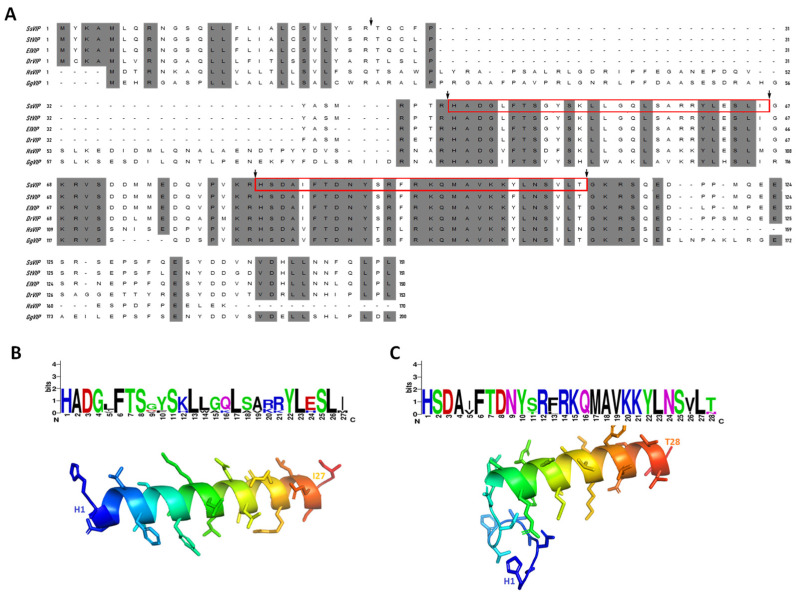
Characterization of prepro-vasointestinal (prepro-VIP) peptide of *S. salar* and secondary structure modeling of mature peptides. (**A**) Amino acid sequence alignment of prepro-VIP of teleost species and other vertebrates. The arrows indicate the putative protease cleavage, and red boxes indicate the peptide-histidine-isoleucine (PHI) and vasoactive intestinal peptide (VIP). (**B**) PyMOL generated the overlapping model based on PHI mature sequence. (**C**) PyMOL generated the overlapping model based on VIP mature sequence.

**Figure 3 animals-12-01174-f003:**
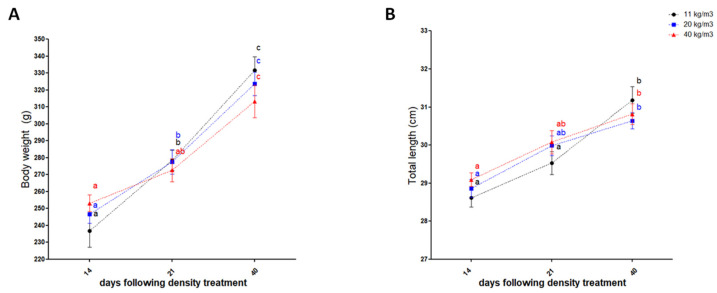
Effect of stocking density on *S. salar* smolt’s growth performance. Values of body weight (**A**) and total length (**B**) of post-smolts cultured at different initial stocking densities after 14, 21, and 40 days are expressed as mean ± SD (*n* = 20). Different letters indicate significant differences between experimental groups (*p* < 0.05, two-way ANOVA, followed by Tukey’s test).

**Figure 4 animals-12-01174-f004:**
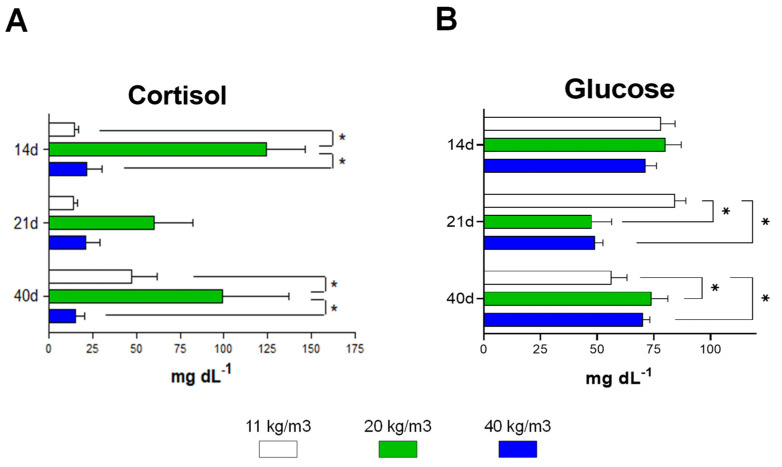
Effect of stocking density plasma biochemical parameters in *S. salar* post-smolts. Values of cortisol (**A**) and glucose (**B**) plasma biochemical parameters were measured after being cultured at different stocking densities after 14, 21, and 40 days. Bars represent mean ± SD (*n* = 4, pools of three fishes). Asterisks show significant differences among stocking densities groups. (*p* < 0.05, two-way, ANOVA, followed by Tukey’s test).

**Figure 5 animals-12-01174-f005:**
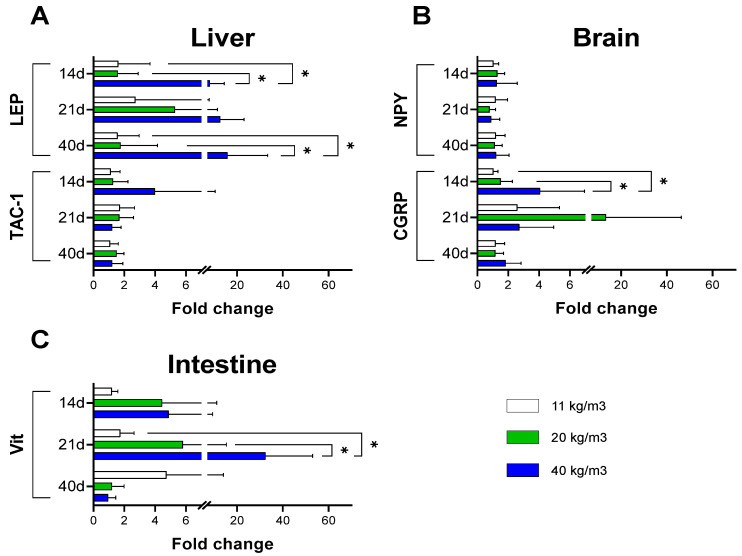
Relative expression of peptide hormones of brain, liver, and intestine of *S. salar* post-smolts cultured at different initial stocking densities. Effect of stocking density on Leptin and TAC-1 mRNA expression in liver on (**A**), NPY and CGRP mRNA expression in brain (**B**), and VIP mRNA expression in intestine (**C**). Bars represent mean ± SD (*n* = 12). Asterisks show significant differences among stocking densities groups (*p* < 0.05, two-way ANOVA, followed by Tukey’s test).

**Figure 6 animals-12-01174-f006:**
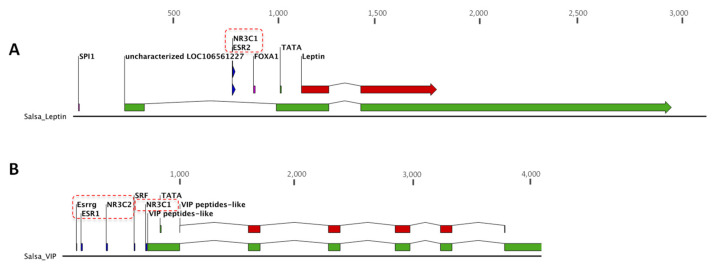
Binding sites of steroid receptor in Lep and VIP genes. The organization of nucleotide sequence of Lep (**A**) and VIP genes (**B**) of Atlantic salmon are shown. The nucleotide region that encodes the mRNA is indicated in green. In the case of VIP mRNA, only a partial 3′-sequence is shown. The red box indicates the different exons of the respective coding sequences. The location of putative sequences corresponding to binding sites recognition of nuclear receptors of steroid hormones is remarked in both nucleotide sequence draws.

## Data Availability

Not applicable.

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
