# Peer review of "Effect of Fish Stock Density on Hormone Genes Expression from Brain and Gastrointestinal Tract of *Salmo salar"

_animals, 2022, doi:10.3390/ani12091174_

Round 1
Reviewer 1 Report
Claudio et al. analyzed the effect of three different stock density on gene expression of NPY, CGRP, Lep, VIP and Prk-1 in Salmo salar post-smolt under circulating aquaculture system. The results showed that the expression of these genes have been upregulated at the high stock density. Meanwhile, the cortisol levels were also increased at the high stock density. These data from this manuscript are very interesting to deserve to publish. I agree with the accept of this manuscript.
Detailed comments
In the section of introduction, what is the stock density of Salmo salar in Chile generally? Let us know why you chose the three different SD.
Line 188-189, how many amplicons have been sequenced? Because it is important for correction of sequences to be analyzed subsequently.
Minor issues
Line 83, “an”→”and”
Line 186-187, this sentence should be rewriten.
Line 449-450, “According to the Wang et al. study” → ”According to the study from Wang et al.”
Author Response
Comments and Suggestions for Authors
Alvarez et al. analyzed the effect of three different stock density on gene expression of NPY, CGRP, Lep, VIP and Prk-1 in Salmo salar post-smolt under circulating aquaculture system. The results showed that the expression of these genes have been upregulated at the high stock density. Meanwhile, the cortisol levels were also increased at the high stock density. These data from this manuscript are very interesting to deserve to publish. I agree with the accept of this manuscript.
Total Response: Thank you very much for your good evaluation. The comments are quite useful for our research. We have revised our manuscript based on your comments.
In the section of introduction, what is the stock density of Salmo salar in Chile generally? Let us know why you chose the three different SD.
Response: In Chile the Atlantic salmon farming densities are over 31 Kg/m3 before the infectious with Salmon Anemia virus (ISAv). Companies justified this as a need to meet the growing demand in international markets. In 2018, the authority issued exempt resolution No. 714/2018, which set maximum farming densities designed to mitigate the occurrence of new outbreaks. This resolution determined the maximum farming densities per species as 11 Kg/m3 for Atlantic salmon and 8 Kg/m3 for Coho salmon and rainbow trout. However, only sanitary reasons were given to define the new culture densities. Here we provide new evidence on the damage to physiological responses that regulate the feed pathway in salmonids, which could be associated with confinement-stress induced by high densities. This new evidence supports the choice of low densities to maintain the animal welfare of these fish under intensive culture.
Line 188-189, how many amplicons have been sequenced? Because it is important for correction of sequences to be analyzed subsequently.
Response: Four different amplicons were analyzed for each gene coding sequence. In addition, platinum Taq High Fidelity polymerase (Invitrogen, Carlsbad, CA) was used for PCR amplification, and both strands of DNA were sequenced. All chromatograms were carefully inspected for sites of ambiguous sequence (double peaks), and those that contained one or more positions of mixed bases were excluded from further analysis. Then, forward and reverse sequence are aligned by ClustalW software, obtaining a consensus sequence. Finally, we traduced DNA to protein sequence to verify the coding sequence.
Minor issues
Line 83, “an”→”and”
Line 186-187, this sentence should be rewriten.
Line 449-450, “According to the Wang et al. study” → ”According to the study from Wang et al.”
Response: These mistakes were corrected in the new version of the manuscript.
Reviewer 2 Report
This is an interesting study that showed how fish stock density regulates hormonal gene expression in the brain and gastrointestinal tract in Samo salar. The manuscript is a good fit to the journal of Animals. I have few concerns that the authors may wish to consider.
Major Concerns:
- Authors may wish to consider title change to “Effect of fish stock density on hormone genes expression in Samo salar”
- Fish size is a major determinant of how fish response to stress this variability will also alter their hormonal levels differently. How did the investigators control for this variable in their study?
Minor concerns:
- Add the SD or SEM of the fish weight
- Be consistent Kg/m3 or Kg/m3
- Oxygen saturation (102.9±3,9) should be (102.9±9)
- Check degree with or without space (4oC or 4 oC) both are correct choose one and stick with it.
- Check figures 5&4 merged with numbering
Author Response
This is an interesting study that showed how fish stock density regulates hormonal gene expression in the brain and gastrointestinal tract in Samo salar. The manuscript is a good fit to the journal of Animals. I have few concerns that the authors may wish to consider.
Total Response: Thank you very much for your good evaluation and your important comments. We listed our responses to your suggestions and comments, on a point-to-point basis.
Major Concerns:
- Authors may wish to consider title change to “Effect of fish stock density on hormone genes expression in Samo salar”
Response: We agree with your suggestion. The new title is more brief and focused on the main results of the study. The new title is then: “Effect of fish stock density on hormone genes expression from brain and gastrointestinal tract of Salmo salar”
- Fish size is a major determinant of how fish response to stress this variability will also alter their hormonal levels differently. How did the investigators control for this variable in their study?
Response: Barcellos et al. 2012 made a significant contribution to the knowledge of the stress, behavior and welfare of fish of different age classes, primarily concerning the timing of measurements and the accurate determination of fish age, regardless of size. For that reason, in this studywe used a smolt batch of the same production cycle, therefore having the same age. This comment was included in the new version of the manuscript in the section Materials and Methods: Fish maintenance and sampling (line 130-133).
Ref: Barcellos LJ, Kreutz LC, Koakoski G, Oliveira TA, da Rosa JG, Fagundes M. Fish age, instead of weight and size, as a determining factor for time course differences in cortisol response to stress. Physiol Behav. 2012 Oct 10;107(3):397-400. doi: 10.1016/j.physbeh.2012.09.008.
Minor concerns:
- Add the SD or SEM of the fish weight
Response: The fish weight was expressed as mean ± SD as indicated in the figure caption.
- Be consistent Kg/m3 or Kg/m3
- Oxygen saturation (102.9±3,9) should be (102.9±9)
- Check degree with or without space (4oC or 4 oC) both are correct choose one and stick with it.
- Check figures 5&4 merged with numbering
Response: These mistakes were corrected in the new version of the manuscript.
Reviewer 3 Report
animals-1666338- “Characterization and stock density effect on the gene expression of peptide hormones from brain and gastrointestinal tract of Salmo salar”
GENERAL COMMENT:
The work entitled “Characterization and stock density effect on the gene expression of peptide hormones from brain and gastrointestinal tract of Salmo salar” is a good work.
This study investigated the effect of varying culture densities of Salmo salar post-smolt on the gene expression of some brain and gastrointestinal hormone signalers.
The subject of the study is very interesting.
Central argument is supported by evidence and analysis.
The methodology described by the author is accurate.
This work is a good work. Discussion section can be expanded and improved, for this reason I require minor revision.
DETAILED COMMENT:
- Title
-The title is adequate.
- Abstract
-In the abstract the objective of the study is clearly described.
Keywords
-In Keywords I suggest adding the scientific name of fish species
- Introduction
-The introduction section is exhaustive.
- Materials and Methods
-This section is accurate
- Results
-This section is accurate and detailed written.
- Discussion
-I suggest improving the section by discussing the results more broadly.
Conclusion
-Conclusion section is sufficient
- Tables and figures
Tables and Figures are clear and understandable
- References
The references are adequate.
Author Response
GENERAL COMMENT:
The work entitled “Characterization and stock density effect on the gene expression of peptide hormones from brain and gastrointestinal tract of Salmo salar” is a good work.
This study investigated the effect of varying culture densities of Salmo salar post-smolt on the gene expression of some brain and gastrointestinal hormone signalers.
The subject of the study is very interesting.
Central argument is supported by evidence and analysis.
The methodology described by the author is accurate.
This work is a good work. Discussion section can be expanded and improved, for this reason I require minor revision.
Total Response: Thank you very much for your positive evaluation and constructive comments.
DETAILED COMMENT:
- Title
-The title is adequate.
- Abstract
-In the abstract the objective of the study is clearly described.
Keywords
-In Keywords I suggest adding the scientific name of fish species
Response: We have added S. salar as a keyword.
- Introduction
-The introduction section is exhaustive.
- Materials and Methods
-This section is accurate
- Results
-This section is accurate and detailed written.
Discussion
-I suggest improving the section by discussing the results more broadly.
Response: Thanks for your valuable suggestion. Three different paragraphs were included in the discussion that joined different results of our study. (line 451-454; 491-496; 535-540).
Conclusion
-Conclusion section is sufficient
- Tables and figures
Tables and Figures are clear and understandable
- References
The references are adequate.